# AgenticIQA: An Agentic Framework for Adaptive and Interpretable Image Quality Assessment

## Abstract

Image quality assessment (IQA) is inherently complex, as it reflects both the quantification and interpretation of perceptual quality rooted in the human visual system. Conventional approaches typically rely on fixed models to output scalar scores, limiting their adaptability to diverse distortions, user-specific queries, and interpretability needs. Furthermore, scoring and interpretation are often treated as independent processes, despite their interdependence: interpretation identifies perceptual degradations, while scoring abstracts them into a compact metric. To address these limitations, we propose **AgenticIQA**, a modular agentic framework that integrates vision-language models (VLMs) with traditional IQA tools in a dynamic, query-aware manner. AgenticIQA decomposes IQA into four subtasks—*distortion detection*, *distortion analysis*, *tool selection*, and *tool execution*—coordinated by a planner, executor, and summarizer. The planner formulates task-specific strategies, the executor collects perceptual evidence via tool invocation, and the summarizer integrates this evidence to produce accurate scores with human-aligned explanations. To support training and evaluation, we introduce **AgenticIQA-200K**, a large-scale instruction dataset tailored for IQA agents, and **AgenticIQA-Eval**, the first benchmark for assessing the planning, execution, and summarization capabilities of VLM-based IQA agents. Extensive experiments across diverse IQA datasets demonstrate that AgenticIQA consistently surpasses strong baselines in both scoring accuracy and explanatory alignment.

## 1 Introduction

Traditional image quality assessment (IQA) aims to quantify perceptual quality in a manner consistent with human vision, serving as a critical bridge between vision science and engineering applications (Duanmu et al., 2021). Existing methods assign scalar scores via full-reference IQA (FR-IQA) models (Wang et al., 2004; Sheikh & Bovik, 2006; Zhang et al., 2018; Ding et al., 2022) or no-reference IQA (NR-IQA) models (Zhang et al., 2020; Zhu et al., 2020; Su et al., 2020; Golestaneh et al., 2022) trained to regress mean opinion scores (MOS). While effective on benchmarks, these models are fundamentally limited: they provide no insight into which distortions influence the score, and operate through rigid pipelines that lack adaptability to varying distortions or user intents.

Recent advances in vision-language models (VLMs) have enabled a complementary approach, framing IQA as a language-driven reasoning task that generates human-aligned explanations (You et al., 2024b; Wu et al., 2024a;c; Chen et al., 2024b; Zhang et al., 2025; Wu et al., 2024b; Zhu et al., 2024). However, VLM-based systems often yield coarse or categorical judgments (*e.g.*, "good" vs. "poor") (Wu et al., 2023a), and remain sensitive to prompt formulations and alignment quality (Li et al., 2025). Furthermore, both traditional and VLM-based IQA systems treat scoring and interpretation as disjoint tasks, despite their inherent interdependence: interpretation reveals the nature of degradation, while scoring compresses this information into a quality model. As shown in Fig. 1(a), such static frameworks either yield accurate but opaque scores or interpretable yet imprecise assessments, limiting their flexibility and utility across diverse IQA tasks.

To address these limitations, we argue that an ideal IQA system should unify the precision of traditional perceptual models with the interpretability and adaptability of VLMs. Specifically, it should: **(i)** adapt its strategy to diverse user queries and visual contexts; **(ii)** leverage calibrated quality scores from full-reference and no-reference IQA tools; and **(iii)** produce transparent, human-aligned explanations alongside its predictions. Realizing this vision requires addressing three key challenges:

- **Adaptivity**: Dynamically generate evaluation strategies based on both image content and user intent, avoiding static pipelines.
- **Modularity**: Seamlessly integrate heterogeneous components, including score-based IQA models and VLM-based IQA models, in a unified framework.

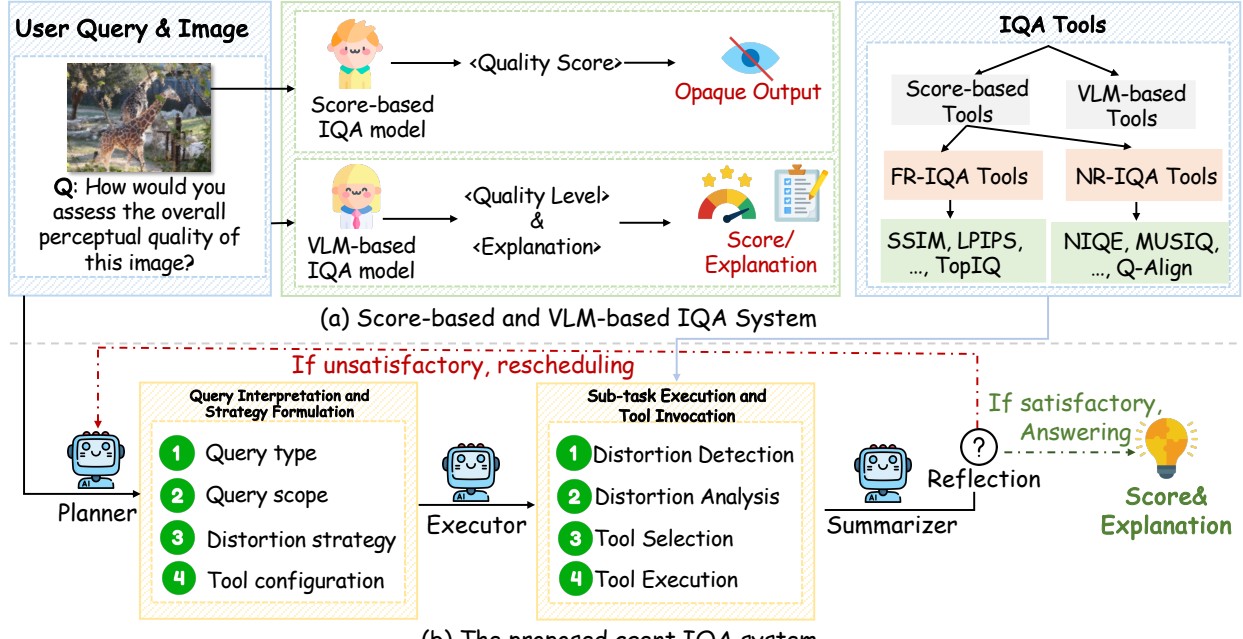

Figure 1: Illustrations of the motivation behind our work. **(a)** Traditional IQA frameworks often rely on a single tool, either a score-based model with accurate but non-explainable outputs or a VLM-based model with interpretable but coarse ratings. Moreover, their static workflows limit adaptability to diverse IQA tasks. **(b)** Our **AgenticIQA** introduces a dynamic agent system that plans and executes IQA sub-tasks based on the user query and image content. It adaptively integrates multi-source quality cues generated during task execution and produces informative, query-aware answers through a refinement process.

- **Interpretability**: Deliver not only accurate quality scores, but also structured, faithful explanations that clarify the rationale behind its decisions.

To this end, we propose **AgenticIQA**, a modular, agent-based IQA framework that decomposes the assessment process into four explicit subtasks: *distortion detection*, *distortion analysis*, *tool selection*, and *tool execution*. These subtasks are orchestrated by a three-agent system using a plan–execute–summarize paradigm (Wang et al., 2023b). As shown in Fig. 1(b), a **planner agent** first generates a query-aware evaluation plan conditioned on the input image and user prompt. An **executor agent** invokes appropriate IQA tools and perceptual detectors to extract structured quality evidence. Finally, a **summarizer agent** integrates the intermediate results to produce an informative, query-specific response, combining scoring and explanation through a refinement process. This design enables *adaptive planning*, *modular integration* of perceptual IQA tools, and *interpretable outputs* via transparent agent behavior. By explicitly separating planning, execution, and reflection, AgenticIQA supports scalable, flexible, and human-aligned visual quality assessment.

In summary, our main contributions are fourfold:

- We propose the **AgenticIQA** framework, the first IQA system to employ planner, executor, and summarizer agents that jointly reason over structured perceptual evidence and language-driven goals.

- We construct **AgenticIQA-200K**, a new large-scale instruction dataset built to train and align VLMs with modular IQA reasoning tasks, supporting plan–execute–summarize learning.

- We introduce **AgenticIQA-Eval**, a benchmark for evaluating VLM-based IQA agents, measuring planning accuracy, execution precision, and summarization reliability through multiple-choice questions (MCQs).

- We demonstrate that **AgenticIQA** consistently outperforms strong traditional and VLM-based baselines across multiple datasets, achieving superior performance in both scoring accuracy and explanation quality.

## 2 RELATED WORK

**Score-based IQA.** Traditional IQA methods are typically categorized as FR-IQA or NR-IQA. FR-IQA applies when a pristine reference image is available and follows either bottom-up or top-down design philosophies. Bottom-up approaches emulate the human visual system (HVS) by incorporating perceptual mechanisms such as contrast sensitivity (Robson, 1966), light adaptation (Boynton & Kandel, 1957), and contrast masking (Legge & Foley, 1980). Top-down approaches rely on high-level HVS assumptions, leading to methods based on structural similarity (Wang et al., 2004; 2003), information theory (Sheikh & Bovik, 2006; Sheikh et al., 2005; Wang & Li, 2011), or deep feature representations (Zhang et al., 2018; Ding et al., 2022; Liao et al., 2022; Zhu et al., 2022). NR-IQA models, though more difficult, are crucial when references are unavailable. They are often informed by assumed distortion types and are divided into distortion-specific and general-purpose methods. Distortion-specific models focus on artifacts introduced by known degradation processes and extract tailored features (Ciancio et al., 2011; Golestaneh & Chandler, 2013; Min et al., 2017). General-purpose models aim for broader applicability, leveraging statistical features (Moorthy & Bovik, 2011; Saad et al., 2012; Mittal et al., 2012) or deep learning (Yang et al., 2022; Su et al., 2020; Ke et al., 2021; Wang et al., 2023a) to capture quality across diverse conditions. Although score-based IQA methods are effective and efficient for standardized scenarios, especially under known distortions, they rely on static evaluation pipelines. By collapsing perceptual quality into a single scalar value, they sacrifice interpretability and offer limited insight into the causes of degradation, reducing their applicability in complex, real-world environments.

**VLM-based IQA.** The emergence of VLMs has opened new avenues for IQA by leveraging their strengths in perceptual reasoning, cross-modal understanding, and natural language generation. Recent efforts have focused on enhancing interpretability (Wu et al., 2024a;c; You et al., 2024b; Chen et al., 2024c; Zhang et al., 2025) and scoring accuracy (Wu et al., 2024b; Zhu et al., 2024; You et al., 2025; Tian et al., 2025; Li et al., 2025). Wu *et al.* pioneered instruction tuning and large-scale human feedback to align foundation models with low-level quality tasks (Wu et al., 2024a;c), while DepictQA introduced a language-driven framework for descriptive assessment (You et al., 2024b). Fine-grained reasoning and distortion localization have been further explored via segmentation-based techniques (Chen et al., 2024b;c). Discrete quality levels, defined textually in both absolute (Wu et al., 2024b; You et al., 2025) and relative (Zhu et al., 2024) forms, have been proposed to align VLM outputs with human perception. Training-free strategies (Liu et al., 2024; Pan et al., 2024) and reinforcement learning frameworks such as GRPO (Li et al., 2025) have also been explored to improve score reliability. Despite their strengths in flexibility and explanation, VLM-based IQA methods often yield imprecise or inconsistent scores, particularly under fine-grained or domain-specific conditions. Their static, single-step reasoning further constrains adaptability in complex evaluation tasks. Therefore, we propose AgenticIQA, a dynamic framework that integrates VLM reasoning with specialized IQA tools through intelligent planning, enabling interpretable, adaptive, and task-specific quality assessment.

**Agent.** The advanced language and planning capabilities of large language models (LLMs) have enabled their integration as core controllers in autonomous agents. Building on these capabilities, LLM-based autonomous agents have been recognized as intelligent entities capable of accomplishing specific tasks, via perceiving the environment, planning, and executing actions (Guo et al., 2024; Huang et al., 2024; Zhao et al., 2024). Recent research has extended the potential of LLM-based agents by introducing role specialization and enabling interactions among multiple agents to simulate complex real-world environments more effectively (Liang et al., 2023; Bo et al., 2024; Wu et al., 2023b; Chan et al., 2023). These advancements have led to successful applications across various domains, including visual tasks such as image restoration (Zhu et al., 2025) and image retrieval (Tu et al., 2025). Inspired by this progress, we propose to utilize agentic reasoning to address the challenges in IQA tasks. Through task decomposition, dynamic planning, and multimodal interaction, the proposed AgenticIQA system orchestrates flexible and interpretable evaluation workflows tailored to diverse user queries and image content.

## 3 AGENTICIQA

The AgenticIQA architecture is organized around three core components, each powered by VLM: the **Planner**, **Executor**, and **Summarizer** (see Fig. 2). These components collaborate in a structured reasoning workflow, with each module fulfilling a distinct role by leveraging the strengths of LLMs. The prompts used in each sub-task can be found in Appendix A.1.

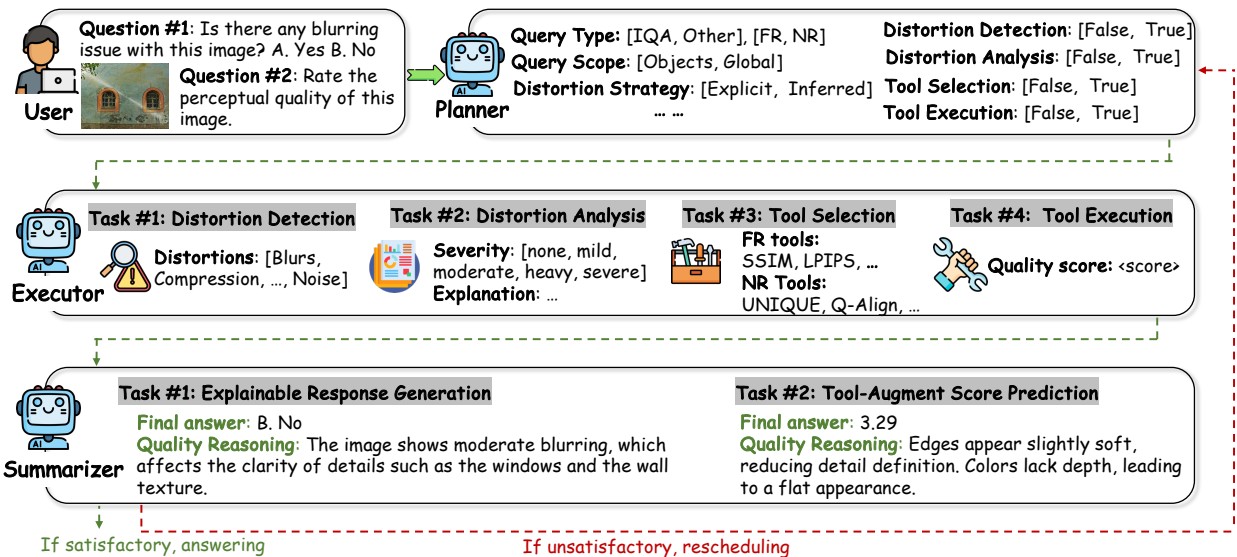

Figure 2: Overview of the **AgenticIQA** framework illustrating the workflow across planner, executor, and summarizer modules.

## 3.1 PLANNER: QUERY INTERPRETATION AND STRATEGY FORMULATION

The reasoning process begins with the construction of a query-aware evaluation plan. Given an input image $x$, an optional reference image $y$, and a user query $t$, the planner module $\mathcal{P}$ acts as a task interpreter, producing a structured plan $P_t$ that guides subsequent assessment steps. The planning procedure consists of four components:

**(1) Query Type.** The planner categorizes the query based on its perceptual focus. Queries related to technical degradations (*e.g.*, noise and blur) are assigned to the "IQA" category, while those concerning high-level properties (*e.g.*, color composition and emotional tone) are categorized as "Other". The planner also determines the reference mode (FR or NR) based on the availability of $y$.

**(2) Query Scope.** If the query implicitly targets specific objects, their names are recorded in a set $O = \{O_1, \ldots, O_m\}$. Otherwise, the planner sets the query scope to "Global".

**(3) Distortion Strategy.** If the query explicitly mentions distortion types, the planner extracts them into a set $D = \{d_1, d_2, \ldots, d_m\}$, sets the *distortion detection* flag to False, and labels the source as "explicit". Otherwise, for open-ended or under-specified queries, it enables *distortion detection* and sets the source to "inferred". *Distortion analysis* is enabled for "IQA" tasks to support interpretability, and disabled for "Other" tasks to minimize unnecessary computation.

**(4) Tool Configuration.** Depending on the query scope (*e.g.*, global vs. local) and any user-specified tool constraints, the planner sets switches for tool selection and execution. If the query is "global" and no specific tool is provided, *tool selection* is enabled; otherwise, it is disabled. The *tool execution* flag is enabled for "global" queries requiring quantitative assessment, and disabled for localized or qualitative reasoning tasks.

## 3.2 EXECUTOR: SUB-TASK EXECUTION AND TOOL INVOCATION

Given the evaluation plan $P_t$, the executor module $\mathcal{E}$ sequentially performs the specified sub-tasks. Each sub-task corresponds to a functional module: *distortion detection* (*dd*), *distortion analysis* (*da*), *tool selection* (*ts*), and *tool execution* (*te*). For each step, the executor consults the control flag in $P_t$ and invokes the associated module only if it is enabled $M_i = \mathcal{E}_i(x, t, P_t(i), \mathcal{T})$, where $i \in \{dd, da, ts, te\}$. Each activated module $\mathcal{E}_i$ produces structured quality cues, which are collectively aggregated into an intermediate multimodal representation $M_t$ used in the subsequent stage.

**(1) Distortion Detection.** When the distortion source is designated as "inferred", the detection module $\mathcal{E}_{dd}$ examines the image $x$ to identify candidate distortion types. The module performs either global or region-based distortion detection to produce a candidate distortion set $D = \mathcal{E}_{dd}(x, t_{dd})$, where $t_{dd}$ is the distortion detection prompt.

**(2) Distortion Analysis.** Given a distortion set $D = \{d_1, \ldots, d_n\}$—either explicitly specified or inferred—the distortion analysis module $\mathcal{E}_{da}$ estimates the severity and perceptual impact of each distortion across image regions or object instances. For each object $O_k \in \{O_1, \ldots, O_m\}$ with associated distortions $D_k \subseteq D$, the module outputs:

$$A_i = \{(d_i, l_i, r_i) \,|\, d_i \in D_k\} = \mathcal{E}_{da}(x, t_{da}, D_k, O_k), \tag{1}$$

where $l_i \in \{\texttt{none}, \texttt{slight}, \texttt{moderate}, \texttt{severe}, \texttt{extreme}\}$ denotes the estimated distortion level, $r_i$ is a concise textual reasoning of its perceptual impact, and $t_{da}$ is the distortion analysis prompt. The full distortion output is constructed as $A = \{A_1, \ldots, A_m\}$ and serves as a structured quality representation for downstream reasoning.

**(3) Tool Selection.** If tool selection is enabled in the evaluation plan, the module $\mathcal{E}_{ts}$ selects an appropriate IQA model for each distortion $d_i \in D$. Each tool in the library $\mathcal{T}$ is annotated with metadata specifying its supported distortion types, concise method descriptions, and reference compatibility. The selected tool is given by:

$$T_i = \mathcal{E}_{ts}(d_i, t_{ts}, \mathcal{T}), \tag{2}$$

where $T_i$ is the selected IQA model for the distortion $d_i$, and $t_{ts}$ is the tool selection prompt

**(4) Tool Execution.** Given a selected tool $T_i$ and input image $x$, the execution module $\mathcal{E}_{te}$ computes a numerical quality score that reflects the severity or perceptual degradation captured by $T_i$:

$$\hat{q}_i = \mathcal{E}_{te}(x, T_i), \tag{3}$$

where $\hat{q}_i \in \mathbb{R}$ denotes the predicted quality score under IQA tool $T_i$. We apply the five-parameter monotonic logistic function to ensure a consistent scoring range across different tools (Sheikh et al., 2006). Details of the IQA tools and the logistic function can be found in Appendix A.3.

The combined outputs from all active modules form the intermediate representation $M_t$, which serves as the foundation for final reasoning and response generation in the summarization stage.

## 3.3 SUMMARIZER: RESPONSE GENERATION AND REFLECTION

The summarizer module $\mathcal{S}$ produces the final response by integrating the intermediate multimodal state $M_t$, which encapsulates perceptual cues collected during execution. It supports both explanation and scoring-oriented queries through structured reflection mechanisms. Before generating the response, the summarizer evaluates whether the collected information in $M_t$ is sufficient to address the query. If so, it synthesizes an answer using the available evidence. Otherwise, it prompts the planner to revise the evaluation strategy, enabling a self-correcting loop for enhanced reliability.

**(1) Explainable Response Generation.** For $\texttt{IQA}$-type queries with global scope, $\mathcal{S}$ synthesizes distortion types, severity levels, and numerical scores to produce a comprehensive answer accompanied by human-aligned justification. For local-object $\texttt{IQA}$ queries, it focuses on region-specific distortion attributes to construct targeted explanations. For $\texttt{Other}$ query types involving aesthetic or semantic cues, the summarizer directly interprets visual content to produce contextually aligned responses.

**(2) Tool-Augment Score Prediction.** To produce a continuous quality score for $\texttt{IQA}$-type queries, the summarizer fuses perceptual signals from multiple IQA tools using a HVS-inspired weighting scheme. Given a set of $n$ tool predictions $\{\hat{q}_i\}_{i=1}^{n}$, where $\hat{q}_i \in [1, 5]$ and lower values indicate perceptual degradation, we first compute the mean predicted quality: $\bar{q} = \frac{1}{n}\sum_{i=1}^{n} \hat{q}_i$. To reflect the nonlinear sensitivity of the HVS, we construct a perceptual weighting vector $\boldsymbol{\alpha} \in \mathbb{R}^5$ across discrete quality levels $c \in \mathcal{C} = \{1, 2, 3, 4, 5\} = \{\texttt{"bad"}, \texttt{"poor"}, \texttt{"fair"}, \texttt{"good"}, \texttt{"excellent"}\}$ using a Gaussian-like function centered at $\bar{q}$:

$$\alpha_c = \frac{\exp(-\eta(\bar{q} - c)^2)}{\sum_{j=1}^{5} \exp(-\eta(\bar{q} - j)^2)}, \tag{4}$$

where $\eta > 0$ controls the sharpness of the decay, and we set $\eta = 1$ in our experiments. In parallel, the summarizer obtains log-probabilities $\log \hat{p}_c$ for each quality level $c$, and converts them into a valid probability distribution:

$$p_c = \frac{\exp(\log \hat{p}_c)}{\sum_{j=1}^{5} \exp(\log \hat{p}_j)}. \tag{5}$$

The final score $q$ is computed as a weighted sum over quality levels: $q = \sum_{c=1}^{\mathcal{C}} \alpha_c \cdot p_c \cdot c$. This approach adaptively emphasizes perceptually salient quality levels based on aggregated tool predictions while incorporating semantic priors from the VLM.

### 3.4 FOUNDATION MODEL

All three **AgenticIQA** components—the planner, executor, and summarizer—run on a shared VLM. The framework supports both proprietary backbones, such as GPT-4o (Hurst et al., 2024), and leading open-source alternatives, such as Qwen2.5-VL (Bai et al., 2025). While the effectiveness of GPT-4o, its closed weights and limited reproducibility hinder large-scale experimentation. As such, we fine-tune Qwen2.5-VL (*QwenLM2.5-7B*) on the **AgenticIQA-200K** corpus to impart task-aligned agentic reasoning, namely Qwen2.5-VL$^*$.

**AgenticIQA-200K Dataset.** To enhance the performance of Qwen2.5-VL-7B, we generate specific instructions tailored to agentic IQA. This dataset, **AgenticIQA-200K**, is constructed to align the model with the structured reasoning demands of planning, execution, and summarization in our framework. Each sample consists of an image-query pair accompanied by a structured task decomposition and corresponding response trace, enabling explicit supervision across agentic sub-tasks. The instruction corpus is organized into three categories: (i) **Planner** instructions, which train the model to interpret the query and construct evaluation strategies; (ii) **Executor** instructions, which guide sub-task execution such as distortion identification, analysis, and tool selection; and (iii) **Summarizer** instructions, which teach response generation based on aggregated perceptual cues. These instructions are automatically generated using GPT-4o (Hurst et al., 2024), drawing upon high-quality IQA reasoning datasets (Q-Pathway (Wu et al., 2024a) and DQ-495K (You et al., 2024a)), and enriched through programmatic augmentation to link perceptual goals, tool usage, and task-specific reasoning patterns. In total, **AgenticIQA-200K** comprises 50K planning, 100K execution, and 50K summarization instruction-response pairs, spanning a wide spectrum of quality degradations, task formulations, and user intents. Additional dataset construction details and schema are provided in the Appendix B.

**Fine-tuning VLMs with Agentic Instructions.** To align Qwen2.5-VL with the structured reasoning requirements of AgenticIQA, we perform full-parameter fine-tuning using the **AgenticIQA-200K** and Q-Instruct-200K (Wu et al., 2024a) instruction corpus. The model is trained jointly across the planner, executor, and summarizer modules using task-specific instruction-response pairs. Training is conducted end-to-end using next-token prediction loss (Bai et al., 2025), allowing the model to learn coherent, context-sensitive responses across planning, execution, and summarization stages. Detailed training settings and hyperparameters are reported in Appendix A.5.

## 4 AGENTICIQA-EVAL BENCHMARK

**Sourcing Diverse Query Types.** To ensure broad coverage of distortion types and assessment scenarios, we curate 500 distorted images from MICBench (Wu et al., 2024c), which includes content from diverse sources with authentic and generative degradations. Additionally, we select 500 pristine images from the Waterloo exploration dataset (Ma et al., 2016) and synthetically degrade them with one or two randomly sampled distortions following the protocol of (You et al., 2024a). The final benchmark includes 750 images for NR-IQA tasks and 250 for FR-IQA tasks. AgenticIQA-Eval is structured into three evaluation tracks: (1) **Planner** (250 samples): evaluates the model's ability to produce subtask configurations and generate valid evaluation plans; (2) **Executor** (500 samples): assesses two core subtasks—distortion identification and severity estimation (250), and distortion-aware tool selection (250); (3) **Summarizer** (250 samples): measures whether the intermediate perceptual state $M_t$ provides sufficient evidence for producing accurate responses.

**Evaluation Protocols.** Each instance is framed as a multiple-choice question (MCQ) with a single ground-truth answer verified by human annotators. Following prior VLM evaluation standards (Wu et al., 2023a), question formats include `What`, `How`, `Which`, and `Yes/No`, reflecting the decision types encountered across the agentic pipeline. Accuracy is used as the primary metric, with each subtrack evaluated independently to isolate component-wise performance. All MCQs undergo manual curation and cross-verification by at least two expert annotators to ensure label consistency and task validity. Additional question samples and annotation protocols are provided in Appendix C.

## 5 EXPERIMENTS

In this section, we first present the experiment settings, including the evaluation benchmarks and baseline competing methods. Next, we present the main results and ablations on AgenticIQA-Eval, IQA datasets (Nikolay et al., 2015; Ciancio et al., 2011; Li et al., 2023), Q-Bench (Wu et al., 2023a). More qualitative comparisons can be found in the Appendix D.

Table 1: Average accuracy (%) of agent-level performance of VLMs within the **AgenticIQA** framework on the **AgenticIQA-Eval** benchmark.

| Model | Planner | Executor | | Summarizer | Overall |
|---|---|---|---|---|---|
| | | *Distortion* | *Tool* | | |
| *Human* | 84.50% | 75.00% | 79.30% | 88.40% | 81.80% |
| mPLUG-Owl3 (Ye et al., 2024a) | 68.00% | 55.20% | 70.40% | 50.80% | 61.10% |
| InternVL2.5 (Chen et al., 2024d) | 76.00% | 49.20% | 73.60% | 77.20% | 69.00% |
| LLaVA-Onevision (Li et al., 2024) | 62.40% | 58.80% | 77.20% | 85.60% | 71.00% |
| Qwen2.5-VL (Bai et al., 2025) | 74.40% | 55.60% | **78.00%** | 84.80% | 73.20% |
| Q-Instruct (Wu et al., 2024a) | 72.40% | 61.60% | 56.80% | 82.00% | 68.20% |
| Q-SiT (Zhang et al., 2025) | 39.20% | 54.80% | 14.00% | 10.40% | 29.60% |
| Qwen2.5-VL$^*$ | 76.80% | 63.20% | 76.40% | 85.20% | 75.40% |
| Claude-3.5-Sonnet (Anthropic, 2024) | 76.80% | 48.40% | 62.00% | 84.80% | 68.00% |
| Gemini-2.0-Flash (Google Cloud, 2024) | 79.20% | 55.60% | 75.60% | 84.40% | 73.70% |
| GPT-4o (Hurst et al., 2024) | **80.40%** | **64.40%** | 74.40% | **86.00%** | **76.30%** |

## 5.1 EXPERIMENTAL SETUPS

**Evaluation Benchmarks.** We evaluate the proposed **AgenticIQA** framework across three complementary settings. First, **AgenticIQA-Eval** assesses agentic reasoning capabilities through structured multiple-choice questions spanning planning, execution, and summarization. Second, for evaluating quality scoring performance, we adopt three representative IQA datasets: **TID2013** (Nikolay et al., 2015), which comprises 24 synthetic distortion types largely unseen during training; **BID** (Ciancio et al., 2011), containing 586 authentically distorted images captured with professional DSLR cameras; and **AGIQA-3K** (Li et al., 2023), featuring generative distortions from advanced text-to-image models that challenge existing NR-IQA methods. Lastly, we include **LLVisionQA** (Wu et al., 2023a), a language-driven benchmark from Q-Bench with $2,990$ image-question pairs targeting low-level perceptual attributes. The questions span Yes/No, What, and How formats, and are organized along two axes—distortion vs. non-distortion and global vs. local perception, enabling comprehensive evaluation of quality-aware visual reasoning.

**Baselines.** We compare AgenticIQA against a comprehensive set of state-of-the-art baselines. These include four general-purpose open-source VLMs: mPLUG-Owl3 (*QwenLM2-7B*)(Ye et al., 2024a), InternVL2.5 (*InternLM2.5-7B*) (Chen et al., 2024d), LLaVA-OneVision (*QwenLM2-7B*) (Li et al., 2024), and Qwen2.5-VL (*QwenLM2.5-7B*) (Bai et al., 2025); two IQA-enhanced VLMs: Q-Instruct (based on mPLUG-Owl2-7B) (Wu et al., 2024a; Ye et al., 2024b) and Q-SiT (based on LLaVA-OneVision-7B) (Zhang et al., 2025; Li et al., 2024); and three proprietary models: Claude-3.5-Sonnet (Anthropic, 2024), Gemini-2.0-Flash (Google Cloud, 2024), and GPT-4o (Hurst et al., 2024). For evaluating quality scoring performance, we compare against four FR-IQA models—LPIPS (Zhang et al., 2018), DISTS (Ding et al., 2022), WaDIQaM (Bosse et al., 2018), and TopIQ (Chen et al., 2024a)—as well as five NR-IQA methods: MUSIQ (Ke et al., 2021), UNIQE (Zhang et al., 2021), TreS (Golestaneh et al., 2022), LIQE (Zhang et al., 2023), and Q-Align (Wu et al., 2024b). Evaluation is based on Spearman's rank correlation coefficient (SRCC) and Pearson linear correlation coefficient (PLCC), measuring the alignment between predicted scores and MOSs.

## 5.2 MAIN RESULTS

**Performance of IQA Agent.** Table 1 reports the agent-level performance of various VLMs within the **AgenticIQA** framework, evaluated across the planner, executor (distortion recognition and tool selection), and summarizer roles. Most general-purpose VLMs exhibit moderate performance in planning and summarization, benefiting from their strong language reasoning capabilities. In contrast, the IQA-enhanced Q-SiT shows severe degradation across all roles, especially in executor and summarizer accuracy, due to its fine-tuning on rigid, fixed-format instruction-response pairs that lack the flexibility needed for adaptive task decomposition and multi-stage reasoning, ultimately limiting its generalization capacity in dynamic IQA settings. In comparison, our proposed Qwen2.5-VL$^*$ achieves notable improvements compared to vanilla Qwen2.5-VL. The overall performance outperforms all open-source baselines and rivals proprietary models such as Gemini-2.0-Flash and Claude-3.5-Sonnet. Although GPT-4o remains the top overall performer, these results highlight the effectiveness of agentic instruction tuning in aligning VLMs with the hierarchical, role-driven demands of perceptual quality assessment.

Table 2: Quality prediction performance of AgenticIQA and both FR-IQA and NR-IQA methods across three standard benchmarks: TID2013 (Nikolay et al., 2015), BID (Ciancio et al., 2011), and AGIQA-3K (Li et al., 2023).

| Method | TID2013 | | BID | | AGIQA-3K | |
|---|---|---|---|---|---|---|
| | SRCC↑ | PLCC↑ | SRCC↑ | PLCC↑ | SRCC↑ | PLCC↑ |
| LPIPS (Zhang et al., 2018) | 0.7445 | 0.7529 | - | - | - | - |
| DISTS (Ding et al., 2022) | 0.8300 | 0.8498 | - | - | - | - |
| WaDIQaM (Bosse et al., 2018) | 0.8058 | 0.8270 | - | - | - | - |
| TopIQ (Chen et al., 2024a) | 0.9075 | 0.9064 | - | - | - | - |
| MUSIQ (Ke et al., 2021) | 0.5750 | 0.6821 | 0.7473 | 0.7701 | 0.6296 | 0.7353 |
| UNIQUE (Zhang et al., 2021) | 0.7507 | 0.7864 | 0.7819 | 0.7801 | 0.6662 | 0.7560 |
| TreS (Golestaneh et al., 2022) | 0.3931 | 0.5444 | 0.6064 | 0.6187 | 0.6493 | 0.7610 |
| LIQE (Zhang et al., 2023) | 0.7982 | 0.8259 | 0.8213 | 0.8192 | 0.7219 | 0.7632 |
| Q-Instruct (Wu et al., 2024a) | 0.6231 | 0.6896 | 0.8670 | 0.8761 | 0.6943 | 0.7776 |
| Q-Align (Wu et al., 2024b) | 0.8313 | 0.8573 | **0.8967** | **0.9151** | **0.8013** | 0.8416 |
| Q-SiT (Zhang et al., 2025) | 0.7686 | 0.8081 | 0.8530 | 0.8656 | 0.7901 | **0.8468** |
| **AgenticIQA** (Qwen2.5-VL*) | 0.7780 | 0.7982 | 0.7771 | 0.8174 | 0.7165 | 0.7967 |
| **AgenticIQA** (GPT-4o) | **0.9165** | **0.9215** | 0.8889 | 0.9093 | 0.7937 | 0.8340 |

Table 3: Average accuracy (%) of the perceptual interpretation MCQs of **AgenticIQA** and the state-of-the-art VLMs on the LLVisionQA test set (Wu et al., 2023a).

| Model | Question Types | Low-level Concerns | Overall |
|---|---|---|---|
| *Junior-level human* | 74.05% | 74.27% | 74.31% |
| *Senior-level human* | 81.76% | 82.52% | 81.74% |
| Q-Instruct | 69.91% | 71.08% | 70.30% |
| Q-SiT | 75.71% | 76.18% | 75.65% |
| mPLUG-Owl3 | 75.31% | 75.32% | 74.21% |
| InternVL2.5 | 74.88% | 74.89% | 73.80% |
| LLaVA-Onevision | 75.68% | 75.68% | 74.68% |
| Qwen2.5-VL | 76.32% | 76.33% | 75.22% |
| Gemini-2.0-Flash | 72.91% | 72.54% | 72.64% |
| Claude-3.5-Sonnet | 72.28% | 72.98% | 72.44% |
| GPT-4o | **78.67%** | 78.67% | 77.88% |
| **AgenticIQA** (Qwen2.5-VL*) | 75.22% | 76.60% | 75.25% |
| **AgenticIQA** (GPT-4o) | 78.11% | **78.76%** | **77.95%** |

Table 4: Comparison of VLM-based IQA and AgenticIQA under identical backbones, highlighting performance gains.

| Dataset | Method | LLaVA-Onevision | Qwen2.5-VL* | GPT-4o |
|---|---|---|---|---|
| TID2013 | VLM-based IQA | 0.6737 | 0.7054 | 0.7567 |
| | **AgenticIQA** | **0.6934** | **0.7780** | **0.9165** |
| | | +0.0197 | +0.0726 | +0.1598 |
| BID | VLM-based IQA | 0.5327 | 0.7674 | 0.8513 |
| | **AgenticIQA** | **0.6845** | **0.7771** | **0.8889** |
| | | +0.1518 | +0.0097 | +0.0376 |
| AGIQA-3K | VLM-based IQA | 0.7515 | 0.7430 | 0.7875 |
| | **AgenticIQA** | **0.7604** | **0.7465** | **0.7937** |
| | | +0.0089 | +0.0035 | +0.0062 |
| LLVisionQA | VLM-based IQA | 74.68% | 75.22% | 77.88% |
| | **AgenticIQA** | **75.02%** | **75.25%** | **77.95%** |
| | | +0.34% | +0.03% | +0.07% |

**Performance of IQA Scoring.** Table 2 presents the quality prediction performance of **AgenticIQA** and a comprehensive set of baseline IQA methods across three standard benchmarks: TID2013, BID, and AGIQA-3K. Traditional FR and NR models such as TopIQ, DISTS, and UNIQUE exhibit strong performance on specific datasets but often struggle to generalize across diverse distortion types and task formulations. Recent VLM-based approaches (*e.g.*, Q-Align and Q-SiT) show improved robustness by incorporating discrete quality-level supervision, yet they remain limited by the resolution of categorical labels and the absence of structured inference. In contrast, **AgenticIQA** (Qwen2.5-VL) achieves competitive zero-shot performance without relying on MOS-based instruction tuning. The **AgenticIQA** (GPT-4o) variant further achieves superior results across all datasets, demonstrating the strength of combining perceptual IQA tools with structured, agentic reasoning. These results underscore the effectiveness of the proposed HVS-inspired weighting scheme and validate the ability to produce accurate quality predictions under diverse conditions.

**Performance of IQA Interpretation.** Table 3 reports the performance of **AgenticIQA** and a range of VLM baselines on the LLVisionQA test set, which evaluates perceptual interpretation through multiple-choice questions covering both query types and low-level quality concerns. **AgenticIQA** (Qwen2.5-VL) achieves a 75.25% overall accuracy,

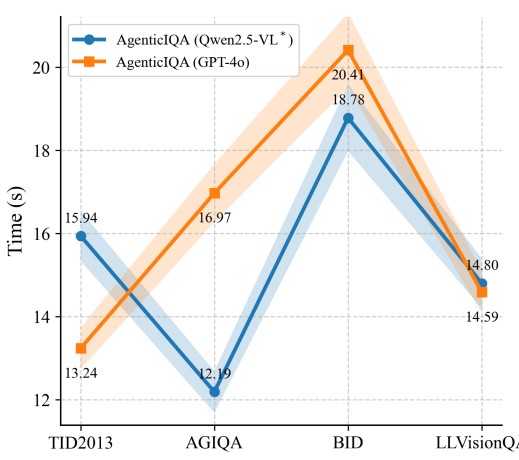
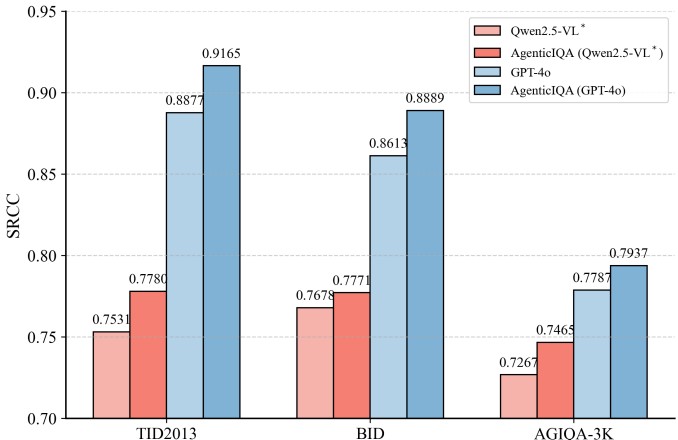

Figure 3: Illustration of average running time per sample on different datasets.

Figure 4: Illustration of comparing the tool-augment score prediction scheme with the uniform averaging.

outperforming its base model (75.22%) and most open-source baselines. Notably, it shows strong gains in perceptual reasoning dimensions such as in-context distortion and object-level interpretation. **AgenticIQA** (GPT-4o) further improves accuracy to 77.95%, surpassing GPT-4o (77.88%) across key categories including `How` questions, `Other` distortions, and in-context reasoning. These results highlight the benefit of structured agentic processing in enhancing local quality perception and language-grounded explanation, particularly under complex and contextual quality queries.

**Running Time Analysis.** We randomly select 50 samples for each benchmark and calculate the average running time per sample. This procedure is repeated five times to ensure reliability, with the final reported results representing the mean across these runs. All experiments are conducted on the same server. As shown in Fig. 3, this increased computational cost primarily results from its multi-agent design, involving structured planning, detailed distortion analysis, and iterative invocation of IQA tools, inherently requiring more computational resources than single-pass evaluation models. Despite its greater computational demands, the proposed AgenticIQA can offer enhanced interpretability, robustness, and adaptability in complex image quality assessment scenarios.

### 5.3 ABLATION STUDIES

**Comparison of VLM-based IQA and AgenticIQA.** As shown in Table 4, AgenticIQA yields notable gains over VLM-based IQA, particularly on challenging settings such as TID2013, where explicit tool selection and execution provide substantial advantages. Improvements are also evident on BID and AGIQA-3K, though more moderate, reflecting the benefit of structured reasoning even under authentic and generative distortions. On LLVisionQA, the enhancements are smaller but consistent, underscoring the robustness of agentic processing in perceptual interpretation tasks. These results confirm that decomposing IQA into modular agentic stages strengthens both scoring accuracy and explanatory alignment beyond what single-pass VLM reasoning can achieve.

**Effect of Different Scoring Schemes.** As shown in Fig. 4, we compare the proposed tool-augmented quality scoring strategy with the widely used uniform averaging approach (Wu et al., 2024b), from which we can observe that our HVS-inspired weighting scheme consistently yields more accurate predictions across all datasets by adaptively emphasizing perceptually salient score levels.

### 6 CONCLUSIONS

We introduced **AgenticIQA**, a modular agent-based framework that unifies traditional perceptual models and VLMs through structured planning, execution, and summarization. By decomposing IQA into interpretable sub-tasks, AgenticIQA enables query-aware evaluation, tool-augmented reasoning, and human-aligned explanations. Extensive experiments show that it consistently surpasses both general-purpose and IQA-enhanced VLMs, including proprietary baselines, underscoring the value of dynamic planning, perceptually grounded execution, and agentic instruction tuning for robust visual quality assessment.

ETHICS STATEMENT

This work focuses on advancing IQA and does not involve the collection or analysis of personally identifiable, sensitive, or harmful data. All datasets used are either publicly available or synthetically generated with appropriate licenses. Human annotations, where applicable, were collected following standard ethical practices, with informed consent obtained and no identifiable personal information retained. The proposed framework is intended solely for academic research and practical quality assessment applications, with no foreseeable misuse toward generating or promoting harmful content.

REPRODUCIBILITY STATEMENT

We ensure reproducibility by providing detailed descriptions of the AgenticIQA framework, benchmark design, and evaluation protocols in the main text and supplementary material. All datasets used are publicly accessible, and the tool libraries integrated into our system are standard open-source IQA models. Experimental settings, hyperparameters, and evaluation metrics are fully specified, and the complete source code is included in the supplementary material to facilitate replication and further research.

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

# A MORE DETAILS ON AGENTICIQA

## A.1 PROMPT TEMPLATES FOR **AGENTICIQA**

Herein, we provide a detailed description of the different prompts used by AgenticIQA for the planner, executor, and summarizer.

## A.2 PROMPTS FOR PLANNER, EXECUTOR AND SUMMARIZER

---

**Prompt for Planner**

**System Message:**
You are a planner in an image quality assessment (IQA) system. Your task is to analyze the user's query and generate a structured plan for downstream assessment.
Return a valid JSON object in the following format:

```
{
"query_type": "IQA" or "Other",
"query_scope": ["<object1>", "<object2>", ...] or "Global",
"distortion_source": "Explict" or "Inferred"
"distortions": dict or null,
"reference_mode": "Full-Reference" or "No-Reference",
"required_tool": list or null,
"plan": {
    "distortion_detection": bool,
    "distortion_analysis": bool,
    "tool_selection": bool,
    "tool_execute": bool
    }
}
```

Instructions:
1. Query Type:
- If the question focuses on visual distortions (e.g., noise, blur, lighting, sharpness), set "query_type": "IQA".
- If the question relates to emotion, style, beauty, or visual appeal, set "query_type": "Other".
2. Query Scope:
- You must extract object or region names from the query if they are mentioned in any form (e.g., "the building", "purple flowers", "the sky", "the subject").
- Set "query_scope" to a list of these object names.
- If no objects or regions are mentioned, then and only then set it to "Global".
3. Distortion Source:
- If the query clearly mentions distortions or visual attributes, such as blur, noise, sharpness, lighting, color, contrast, saturation, brightness, etc., set "distortion_source" to "explicit". Otherwise, set it to "inferred".
4. Distortions:
- If the query refers to any specific distortions, must set "distortions" to a dictionary with object names/global as keys and lists of distortions as values. If no distortions are mentioned, set it to null.
5. Reference Mode:
- If both distorted and reference images are present, set "reference_Mode" to "Full-Reference". Otherwise, set to "No-Reference".
6. Required Tools:
- ONLY include tool names if they are explicitly mentioned by name in the user's query.
7. Plan:
- If the query is NOT an IQA task, set all steps (distortion_detection, tool_selection, distortion_analysis, tool_execute) to false. - Set "distortion_detection" to false if any distortions are explicitly mentioned in the query. Otherwise, distortion_detection=true.
- Set "distortion_analysis" to true by default.
- Set "tool_selection" and "tool_execute" according to whether tools/regions are explicitly mentioned:
- If both tool and region are given: tool_selection = false, tool_execute = true.
- If region but no tool: tool_selection = false, tool_execute = false.

---

- If tool but no region: tool_selection = false, tool_execute = true.
- If neither: tool_selection = true, tool_execute = true.

**User Message:**
User's query: {query}

---

**Prompt for Executor (Distortion Detection)**

**System Message:**
You are an expert in distortion detection. Based on the user's query, identify all possible distortions need to be focused on to properly address the user's intent.
Return a valid JSON object in the following format:

```
{{
"distortion_set": {{
    <object_name or "Global">: [<distortion_1>, <distortion_2>,...]
    }}
}}
```

Instructions:
1. Focus your analysis on query scope. Describe distortions for each individually.
2. Only include distortion types from the following valid categories:[ "Blurs", "Color distortions", "Compression", "Noise", "Brightness change", "Sharpness", "Contrast" ]

**User Message:**
User's query: {query}
The image: <image>

---

**Prompt for Executor (Distortion Analysis)**

**System Message:**
You are a distortion analysis expert. Your task is to assess the severity and visual impact of various distortion types for different regions of an image or the entire image.
The distortion information: {distortion_set}
Return a valid JSON object in the following format:

```
{{
"distortion_analysis": {{
<object_name or "Global">: [
            {{
                "type": "<distortion_1>",
                "severity": "<none/slight/moderate/severe/extreme>",
                "explanation": "<brief visual explanation>"
            }},
            ...
            ]
        }}
}}
```

Instructions:
1. Base your analysis on the listed distortion types and consider the user question.
2. Use "none" if a distortion is barely or not visible.
3. Keep explanations short and focused on visual quality. Focus solely on analyzing visual distortion effects.

**User Message:**
User's query: {query}
The image: <Image>

**Prompt for Executor (Tool Selection)**

**System Message:**
You are a tool executor. Your task is to assign the most appropriate IQA tool to each visual distortion type, based on the descriptions of the tools.
The distortion information: {distortion_set}.
The available tools: {tool description}.
Return a valid JSON object in the following format:

```
{
"selected_tools": {
    <object_name or "Global">: {
        <distortion_1>: <tool_1>, <distortion_2>: <tool_2>}
    }
}
```

Instructions:
For each distortion, choose the tool whose description suggests it performs best for that type of distortion.

**User Message:**
User's query: {query}

---

**Prompt for Summarizer (Visual Quality Interpretation)**

**System Message:**
You are a visual quality assessment assistant. Your task is to select the most appropriate answer to the user's question. You are given:
- Distortion analysis (severity and visual impact of listed distortions)
- Tool response (overall quality scores from IQA models)
- Image content

Decision process
1. First, understand what kind of visual information is needed to answer the user's question.
2. Check if the provided distortion analysis or tool response already contains the required information.
3. If the provided information is sufficient, use it to answer.
4. If the information is unclear or insufficient, analyze the image directly to determine the best answer.

Return a valid JSON object in the following format:

```
{
"final_answer": "<one of the above letters>",
"quality_reasoning": "<brief explanation, based on either distortion
analysis, tool response, or direct visual observation>"
}
```

Instructions:
For each distortion, choose the tool whose description suggests it performs best for that type of distortion.

**User Message:**
User's query: query The image: <Image>

---

**Prompt for Summarizer (Quality Score Prediction)**

**System Message:**

You are a visual quality assessment assistant. Given the question and the analysis (tool scores, distortion analysis). Your task is to assess the image quality.
You must select one single answer from the following:

A. Excellent
B. Good
C. Fair
D. Poor
E. Bad

**User Message:**
User's query: query
The image: <Image>

## A.3 TOOL DESCRIPTION

A diverse set of IQA tools is integrated into AgenticIQA to support comprehensive visual quality assessment across various scenarios, including full-reference and no-reference evaluation tasks. All tools are sourced from the well-established IQA-PyTorch library [1], which provides standardized implementations. The tool selection is guided by their effectiveness in handling different distortion types and alignment with human perception in quality prediction. To evaluate the strength of each tool in handling different distortions, we conduct performance analysis on the KADID-10k dataset (Lin et al., 2019), which provides a rich set of 25 controlled distortion types with varying levels. The detailed descriptions of the tools are provided below.

- **TOPIQ (Chen et al., 2024a):** A top-down FR-IQA model that leverages high-level semantic guidance to focus on perceptually important distortion regions, thereby enhancing assessment accuracy. Best at evaluating: - Blurs (lens blur, motion blur) - Color distortions (color diffusion, color shift, color quantization, color saturation) - Compression (JPEG2000 and JPEG) - Noise (white noise, color component noise, impulse noise, multiplicative noise, denoise artifact) - Brightness change (brighten, darken, mean shift) - Spatial distortions (jitter, non-eccentricity patch, pixelate, quantization, color block) - Sharpness and contrast quantization, color block), and contrast/sharpness variations.

- **AHIQ (Lao et al., 2022):** An attention-guided FR-IQA model tailored to assess distortions commonly introduced by generative models (*e.g.*, GANs). It integrates hybrid mechanisms to improve robustness under complex generation artifacts. This tool has no known strengths for any specific distortion.

- **FSIM (Zhang et al., 2011):** A widely used FR-IQA model based on low-level feature similarity, such as phase congruency and gradient magnitude. This tool has no known strengths for any specific distortion.

- **LPIPS (Zhang et al., 2018):** A deep feature-based FR-IQA metric that computes perceptual similarity aligned with human visual judgments. This tool has no known strengths for any specific distortion.

- **DISTS (Ding et al., 2022):** A structural-texture hybrid similarity model that balances sensitivity to structural degradations and tolerance to textural variations. Best at evaluating: - Blurs (Gaussian blur).

- **WaDIQaM_FR (Bosse et al., 2018):** A Siamese-network-based FR-IQA framework that applies weighted average pooling to fuse predictions from reference and distorted images for quality estimation. This tool has no known strengths for any specific distortion.

- **PieAPP (Prashnani et al., 2018):** A pairwise preference-based FR-IQA model that learns perceptual differences directly from human annotations. Designed to align with subjective quality judgments. This tool has no known strengths for any specific distortion.

- **MS-SSIM (Wang et al., 2003):** An extension of SSIM that computes multi-scale structural similarity, providing a more comprehensive account of image structure across resolutions. This tool has no known strengths for any specific distortion.

- **GMSD (Xue et al., 2013):** Measures image quality by capturing local gradient magnitude deviations. Particularly effective in detecting visually important structural distortions. This tool has no known strengths for any specific distortion.

- **SSIM (Wang et al., 2004):** A foundational FR-IQA model based on luminance, contrast, and structure comparisons. Widely used in denoising, deblurring, and super-resolution evaluations. This tool has no known strengths for any specific distortion.

---

[1] https://github.com/chaofengc/IQA-PyTorch

- **CKDN (Zheng et al., 2021):** A knowledge-distillation-based FR-IQA model that incorporates degraded reference images to improve robustness under partial-reference conditions. This tool has no known strengths for any specific distortion.

- **VIF (Sheikh & Bovik, 2006):** Evaluates quality based on the amount of visual information retained between reference and distorted images. It ranks highly for JPEG and JPEG2000 compression. This tool has no known strengths for any specific distortion.

- **PSNR:** A classical pixel-wise metric that computes the logarithmic ratio of peak signal power to distortion noise. Still prevalent in compression and restoration tasks.

- **VSI (Zhang et al., 2014):** Integrates visual saliency into FR-IQA by emphasizing regions likely to draw human attention. It provides enhanced perceptual alignment for saliency-sensitive distortions. This tool has no known strengths for any specific distortion.

- **QAlign (Wu et al., 2024b):** A state-of-the-art NR-IQA model based on multimodal large language models (MLLMs). Best at evaluating: - Blurs (Gaussian blur, motion blur) - Color distortions (color shift, color quantization, color saturation) - Noise (white noise, color component noise, impulse noise, multiplicative noise) - Brightness change (brighten, darken, mean shift) - Spatial distortions (jitter, quantization) - Sharpness.

- **CLIPIQA (Wang et al., 2023a):** An NR-IQA method that leverages CLIP embeddings to measure semantic fidelity and perceptual degradation. This tool has no known strengths for any specific distortion.

- **UNIQIE (Zhang et al., 2021):** An uncertainty-aware NR-IQA model designed to estimate quality under both synthetic and real-world degradations. Best at evaluating: - Blurs (lens blur) - Compression (JPEG, JPEG2000) - Noise (denoise artifact) - Spatial distortions (non-eccentricity patch, pixelate, color block) - Contrast.

- **HyperIQA (Su et al., 2020):** A self-adaptive architecture that decouples IQA into content understanding, perceptual rule learning, and score prediction, enabling flexible generalization across diverse image contexts. This tool has no known strengths for any specific distortion.

- **TReS (Golestaneh et al., 2022):** A transformer-based blind IQA model that incorporates relative ranking and consistency learning to capture both global and local perceptual features. This tool has no known strengths for any specific distortion.

- **MUSIQ (Ke et al., 2021):** A multi-scale transformer that processes images at native resolutions and varying aspect ratios, offering robust perceptual quality prediction via hierarchical feature fusion. This tool has no known strengths for any specific distortion.

- **WaDIQaM_NR (Bosse et al., 2018):** Applies a deep neural network with weighted average pooling to perform NR-IQA by aggregating spatially varying local quality scores. This tool has no known strengths for any specific distortion.

- **DBCNN (Zhang et al., 2020):** A bilinear CNN architecture that extracts and fuses local-global representations for no-reference quality prediction. This tool has no known strengths for any specific distortion.

- **ARNIQA (Agnolucci et al., 2024):** A self-supervised NR-IQA model that learns a distortion manifold for quality representation, facilitating robust prediction without reference supervision. This tool has no known strengths for any specific distortion.

- **NIMA (Talebi & Milanfar, 2018):** Predicts aesthetic and technical quality using probability distributions derived from human ratings. Widely used for aesthetic assessment tasks. This tool has no known strengths for any specific distortion.

- **BRISQUE (Mittal et al., 2012):** A pioneering NSS-based NR-IQA model that captures spatial statistical deviations in natural images. This tool has no known strengths for any specific distortion.

- **NIQE (Mittal et al., 2013):** An opinion-unaware metric based on statistical regularities in natural images. This tool has no known strengths for any specific distortion.

- **MANIQA (Yang et al., 2022):** Combines visual transformers with quality-aware attention mechanisms to evaluate GAN-generated distortions and other complex artifacts. This tool has no known strengths for any specific distortion.

- **LIQE (Zhang et al., 2023):** A multitask-learning-based blind IQA model that exploits auxiliary tasks to enhance distortion awareness. Best at evaluating: - Color distortion (color diffusion).

## A.4 UNIFIED SCORING STRATEGY FOR TOOLS

To ensure score consistency across heterogeneous IQA tools and enable fair comparison, we apply a five-parameter monotonic logistic transformation (Sheikh et al., 2006) to normalize the predicted quality scores. The parameters $\{\beta_i\}_{i=1}^5$ are fitted on the KADID-10k dataset (Lin et al., 2019) to align each model's outputs onto a comparable scale (*i.e.*, [1, 5], larger value indicates better visual quality). Following the standard form (Sheikh et al., 2006), the transformed score $\tilde{q}_i$ is computed as:

$$\hat{q}_i = \beta_1 \left( \frac{1}{2} - \frac{1}{\exp(\beta_2(\tilde{q}_i - \beta_3))} \right) + \beta_4 \tilde{q}_i + \beta_5, \tag{6}$$

where $\tilde{q}_i$ denotes the raw score predicted by the tool. These aligned scores $\hat{q}_i$ are used in downstream evaluations to mitigate discrepancies due to varying tool-specific output distributions. Table 5 summarizes the fitted parameters for each full-reference and no-reference IQA model.

Table 5: Fitted parameters of the five-parameter logistic function used to align the score ranges of different IQA models. Parameters are estimated on the KADID-10k dataset to enable unified evaluation.

| IQA model | $\beta_1$ | $\beta_2$ | $\beta_3$ | $\beta_4$ | $\beta_5$ |
|---|---|---|---|---|---|
| TopIQ_FR | 21.73 | 0.1147 | 0.4721 | 3.5654 | 1.0094 |
| AHIQ | 0.4280 | -592.6356 | -2.4819 | 3.7677 | 1.6143 |
| FSIM | 265.7031 | 23.2940 | 1.2003 | 1.9193 | 133.0061 |
| LPIPS | -2.0915 | 3.7543 | -28.0133 | -4.7251 | 4.9710 |
| DISTS | -6.3380 | -7.0362 | −147.7936 | -8.4514 | 1.0753 |
| WaDIQaM_FR | 41.8259 | 0.0997 | -0.3064 | 23.1826 | 3.5943 |
| GMSD | -5.9925 | -23.3876 | -59.6895 | -13.8274 | 1.0789 |
| SSIM | 94.4202 | 64.9155 | 1.0664 | 2.8744 | 47.6819 |
| CKDN | 21.6375 | 3.1301 | 0.3708 | -12.3232 | 6.7834 |
| VIF | 0.4119 | 49.7978 | 0.2237 | 2.4370 | 1.3850 |
| QAlign | 28.8204 | 0.1469 | 6.1941 | -0.1906 | 6.7863 |
| CLIPIQA | 21.7287 | 0.1147 | 0.4721 | 3.5654 | 1.0094 |
| UNIQUE | 52.5605 | 0.2967 | 0.8558 | -2.9510 | 5.6368 |
| HyperIQA | 0.4071 | 39.0359 | 0.3155 | 3.1472 | 0.9815 |
| TReS | -0.0550 | 6559.4161 | 48.9070 | 0.0255 | 1.1176 |
| MUSIQ | 2.4078 | -0.1123 | 32.0686 | 0.0734 | -0.7363 |
| WaDIQaM_NR | 106.9844 | 1.2931 | -0.3321 | -31.9917 | -8.0786 |
| DBCNN | 38.5349 | 0.0851 | 0.5159 | 3.4921 | 0.8607 |
| ARNIQA | 2.2932 | -13.4107 | 0.2884 | 7.5144 | -0.6274 |
| NIMA | 1.0129 | 5.3579 | 4.6475 | 0.3207 | 1.0601 |
| BRISQUE | -2.2106 | 0.0684 | 54.3418 | 0.0050 | 2.2728 |
| NIQE | -1.4174 | 0.8785 | 6.9416 | -0.0059 | 2.7374 |
| MANIQA | 0.6818 | 27.4817 | 0.2621 | 2.5196 | 1.5758 |
| LIQE | 0.1494 | 7.1114 | 3.2801 | 0.6936 | 0.8655 |

## A.5 IMPLEMENTATION DETAILS

We fine-tune the cutting-edge **Qwen2.5-VL** (Bai et al., 2025), an open-source multimodal model that couples a CLIP-ViT-L14 vision encoder (Radford et al., 2021) with the Qwen2.5-7B language decoder, using the proposed **AgenticIQA-200K** and Q-Instruct-200K (Wu et al., 2024a) dataset. The Q-Instruct-200K dataset comprises 200K instruction-response pairs designed for IQA, including 58K explainable pathway reasoning samples, 133K visual question answering examples (76K `What/How` and `57K yes/no`), and 12K `extended` conversations. The training process follows the official implementation[2], with a batch size of 512 distributed across four NVIDIA A100 GPUs (80GB each). We employ a cosine learning rate schedule with an initial learning rate of $1 \times 10^{-5}$ and optimize the model using the next-token prediction loss for two epochs. During inference, a single NVIDIA RTX 3090 GPU suffices to execute the full AgenticIQA pipeline, including the planner, executor, and summarizer modules.

---

[2]https://github.com/QwenLM/Qwen2.5-VL

## B    MORE DETAILS ON AGENTICIQA-200K

In this section, we provide an in-depth overview of the construction process and underlying rationale for the AgenticIQA-200K dataset.

**Image Collection.**    Inspired by successful annotation-free dataset construction strategies (Wu et al., 2024c), we leverage reliable information from existing high-quality IQA datasets, specifically Q-Pathway (Wu et al., 2024a) and DQ-495K (You et al., 2024a), alongside advanced proprietary models (GPT-4o (Hurst et al., 2024)). Specifically, we gather $55,620$ synthetically distorted images from DQ-495K (You et al., 2024a), characterized by carefully documented artificial distortions, and $10,797$ authentically distorted images from Q-Pathway (Wu et al., 2024a), representative of real-world degradations. Notably, each image selected for inclusion is paired with comprehensive perceptual reasoning annotations, capturing detailed distortion attributes and visual quality insights.

**Query Generation.**    Utilizing the detailed perceptual reasoning descriptions accompanying each image, we instruct GPT-4o to systematically transform these annotations into structured IQA-related question-response pairs (Wu et al., 2024a). In total, we obtain approximately 140K question-response pairs, categorized into three question types for comprehensive coverage: (i) 60K `What/How/Which` questions designed to elicit descriptive reasoning about distortions or quality aspects, (ii) 40K `Yes/No` questions aimed at explicit binary evaluations regarding image quality or distortion visibility, and (iii) 40K `Extended` questions that demand deeper reasoning involving comparative analyses, model-based explanations, or conditional quality assessments.

The generated queries are meticulously designed to cover diverse aspects of image quality evaluation, including global and localized distortions, perceptual characteristics of specific objects or scenes, comparative quality assessments, and detailed discussions on the applicability and reliability of existing IQA models. Moreover, each question-response pair is aligned explicitly with the structured agentic tasks (planning, execution, summarization), thus ensuring targeted supervision for training modular IQA agents.

**Instruction and Trace Generation.**    Following query generation, each image-query pair is further augmented with structured task decompositions and explicit response traces, corresponding directly to the agentic sub-task structure in our framework. Specifically, GPT-4o is guided by programmatically designed prompting schemas that explicitly instruct it to decompose each query into a sequential task plan, select relevant IQA tools, and generate execution traces aligned with distinct subtasks. These augmented annotations enable fine-grained, interpretable supervision across each agentic module:

- **Planner Instructions**: These instructions guide the model in query interpretation and formulation of evaluation strategies, including recognizing query intent, identifying necessary sub-tasks, and defining optimal execution sequences.

- **Executor Instructions**: These instructions focus explicitly on operationalizing sub-tasks such as distortion detection, quality analysis, and automated IQA tool selection based on perceptual reasoning outcomes.

- **Summarizer Instructions**: These instructions teach the model to synthesize and generate coherent, insightful summaries that integrate multiple perceptual evaluations into unified, human-interpretable quality assessments.

**Data Filtering and Balancing.**    To guarantee dataset reliability and instructional coherence, we implement filtering criteria, systematically removing apparent incorrect and irrelevant responses generated by GPT-4o. This rigorous filtering process results in a curated dataset with balanced instructional categories, comprising precisely 50K planner, 100K executor, and 50K summarizer instruction-response pairs. Each subset covers diverse distortion types, perceptual evaluation tasks, and user queries, thus enhancing the robustness and generalization capabilities of agentic IQA systems trained on this dataset.

Overall, AgenticIQA-200K provides a structured, comprehensive, and high-quality instructional dataset explicitly designed to facilitate advanced modular reasoning and robust image quality assessment capabilities. The prompts utilized for generating queries and creating planner, executor, and summarizer instructions are detailed as follows.

---

**Prompt for Query Generation**

**System Message:**
You are a multimodal LLM trained for Image Quality Assessment (IQA). Your task is to generate diverse question-answer pairs based on the given image quality description, covering both global perception and local

---

in-context analysis. The questions should be designed to improve the reasoning, perception, and judgment capabilities of an IQA agent.

**Return a valid JSON list in the following format:**

```
[
  {
    "question": str,
    "options": List[str] or None,
    "answer": str
  },
  ...
]
```

**Instructions:**

1. Generate different types of questions:
   - **What / How / Why:** Three multi-choice questions starting with "What", "How", and "Why". Each should include four answer options (A. B. C. D.), where one is correct and others are plausible but incorrect.
   - **Yes-No:** Two binary questions answered with "Yes" or "No", balanced in polarity. Use global or local perceptual judgments and mention distortions or tools where appropriate.
   - **Extended:** Two open-ended questions that require comprehensive answers, covering aspects such as:
     - Global quality summary (e.g., Poor/Bad/Fair/Good/Excellent)
     - Local object distortions and their impact
     - Causes of distortions
     - Suggestions for improvement
     - Tools suitable for assessment

2. At least one question must involve a **specific distortion** (e.g., blur, noise, compression) and one must reference an **IQA tool**, chosen from either:
   - **FR-IQA tools:** TOPIQ, AHIQ, FSIM, LPIPS, DISTS, WaDIQaM_FR, PieAPP, MS-SSIM, GMSD, SSIM, CKDN, VIF, PSNR, VSI
   - **NR-IQA tools:** QAlign, CLIPIQA, UNIQIE, HyperIQA, TReS, MUSIQ, WaDIQaM_NR, DBCNN, ARNIQA, NIMA, BRISQUE, NIQE, MANIQA, LIQE

3. Ensure diversity across:
   - **Scope:** Global perception vs. local object-specific perception
   - **Task Type:** Descriptive, analytic, causal, comparative, or judgmental
   - **Tool Use:** Questions involving appropriate tool selection or suitability

4. Do not output any additional text. Only return a valid JSON list.

**User Message:**
Image Quality Description: {description}

---

**Prompt for Planner Instruction**

**System Message:**
You are a Planner in an Image Quality Assessment (IQA) agent system. Your task is to analyze the user's query and generate a structured plan for downstream assessment. Please follow the instructions below.

**Return a valid JSON list in the following format:**

```
{
  "task_type": "IQA" or "Other",
  "reference_type": "Full-Reference" or "No-Reference",
  "required_object_names": ["<object1>", ...] or null,
  "required_distortions": {"<object_name>" or "Global":
  ["<distortion1>", ...]} or null,
```

```
    "required_tools": ["<tool_name1>", ...] or null,
    "distortion_source": "explicit" or "inferred",
    "plan": {
      "distortion_detection": true or false,
      "tool_selection": true or false,
      "distortion_analysis": true or false,
      "tool_execute": true or false
    }
}
```

**Instructions:**

1. **Task Type:**
   - If the query concerns image quality assessment, set `"task_type"` to `"IQA"`.
   - Otherwise, set it to `"Other"`.

2. **Reference Type:**
   - If both distorted and reference images are mentioned, set `"reference_type"` to `"Full-Reference"`.
   - Otherwise, set it to `"No-Reference"`.

3. **Required Object Names:**
   - Extract object/region names (e.g., "the building", "purple flowers") from the query.
   - If none are found, set to `null`.

4. **Required Distortions:**
   - If distortions are tied to regions, use those region names as dictionary keys.
   - If distortions apply to the whole image, use `"Global"` as the key.
   - If no distortions are referenced, set to `null`.
   - Map descriptive terms to standard categories:
     - "saturation", "colorful", "vivid" → `Colorfulness`
     - "sharp", "blurry", "compression", "JPEG" → `Sharpness`
     - "dark", "bright", "lighting", "exposure" → `Brightness`
     - "contrast" → `Contrast`
     - "noise", "noisy" → `Noise`

5. **Required Tools:**
   - Include only if specific tool names are explicitly mentioned in the query (e.g., "use LPIPS").
   - Do not infer or recommend tools; if none mentioned, set to `null`.

6. **Distortion Source:**
   - If distortion-related terms are mentioned, set to `"explicit"`.
   - Otherwise, use `"inferred"`.

7. **Plan:**
   - If `"task_type"` is `"Other"`, set all flags to `false`.
   - If distortions are mentioned, set `"distortion_detection"` to `false`; else `true`.
   - Always set `"distortion_analysis"` to `true`.
   - If both region and tool are mentioned: `"tool_selection"` = `false`, `"tool_execute"` = `true`.
   - If only region is mentioned: `"tool_selection"` = `false`, `"tool_execute"` = `false`.
   - If only tool is mentioned: `"tool_selection"` = `false`, `"tool_execute"` = `true`.
   - If neither is mentioned: `"tool_selection"` = `true`, `"tool_execute"` = `true`.

**User Message:**
User Query: {query}
```

**Prompt for Executor Instruction (Distortion Analysis) Instruction**

**System Message:**
You are a **distortion detector and visual quality analyzer** in an Image Quality Assessment (IQA) agent system. Your task is to assess the visual distortions in the image and describe their severity and perceptual impact to help answer the user query.

**Return a valid JSON list in the following format:**

```
[
  {
    "distortion_analysis": {
      "<object_name or 'global'>": [
        {
          "type": "<distortion_name>",
          "severity": "<none/mild/moderate/heavy/severe>",
          "explanation": "<brief visual explanation>"
        }
      ]
    }
  }
]
```

**Instructions:**

1. If specific object names or regions are mentioned, focus analysis on those; otherwise, assess the entire image.

2. **Reference Type:**
   - If both distorted and reference images are available, compare them directly.
   - Otherwise, infer distortions from visual cues in the distorted image alone.

3. Identify up to **two** clearly visible distortions per region/object.

4. For each distortion, assign a severity label based on perceptual cues:
   - $1 \rightarrow$ "none"
   - $2 \rightarrow$ "mild"
   - $3 \rightarrow$ "moderate"
   - $4 \rightarrow$ "heavy"
   - $5 \rightarrow$ "severe"

5. Use visual evidence such as texture loss, color shifts, noise intensity, edge clarity, over-/under-exposure.

6. Only include distortion types from the following categories: `Blurs`, `Color distortions`, `Compression`, `Noise`, `Brightness change`, `Sharpness`, `Contrast`

7. If no visible distortions are present, return an empty list.

8. **Do not include extra fields, comments, markdown, or explanations. Return JSON only.**

**Optional Additional Instructions (based on context):**

- If distortions are **explicitly stated** in the plan, refer to those and assess their visual impact accordingly.
- If distortions are **inferred**, reason based on observed cues. Optionally, leverage the provided quality description for additional guidance.

**User Query:** {query}

**Prompt for Executor (Tool Assignment) Instruction**

**System Message:**
You are a **tool executor** in an Image Quality Assessment (IQA) agent system. Your task is to assign the most appropriate IQA tool to each identified visual distortion, based on the tool descriptions and the distortion analysis below. Your goal is to select the best-matched tool to support high-accuracy quality scoring.

**Return a valid JSON list in the following format:**

```
{
  "selected_tools": {
    "<object_name or 'global'>": {
      "<distortion_name_1>": "<tool_name>",
      "<distortion_name_2>": "<tool_name>"
    }
  }
}
```

**Instructions:**

1. If **required tools** are already specified by the planner, you must directly use them for the corresponding distortions.

2. Otherwise, select tools based on the following principles:
   - Choose the tool whose description indicates high performance on that distortion type.
   - If no exact match, choose the tool most semantically related to the distortion.

3. Tool names must exactly match the provided list. Do **not** add any prefixes, suffixes, or formatting (e.g., do not write `function.LPIPS`).

4. Return **only valid JSON**. Do not include explanations, markdown, or commentary in your response.

**Available Tools:**

- **FR-IQA tools:** TOPIQ_FR, AHIQ, FSIM, LPIPS, DISTS, WaDIQaM_FR, PieAPP, MS-SSIM, GMSD, SSIM, CKDN, VIF, PSNR, VSI

- **NR-IQA tools:** QAlign, CLIPIQA, UNIQIE, HyperIQA, TReS, MUSIQ, WaDIQaM_NR, DBCNN, ARNIQA, NIMA, BRISQUE, NIQE, MANIQA, LIQE

**Optional Tool Filtering:** If the planner specifies a list of allowed tools, you must choose only from that subset.

**Distortion Analysis Example:**

```
Object: sky
- Distortion: color shift
  - Severity: moderate
  - Explanation: noticeable unnatural tint across blue regions

Object: tree
- Distortion: blur
  - Severity: heavy
  - Explanation: edges are smoothed and details are lost
```

**User Query:** {query}

---

**Prompt for Summarizer Instruction**

**System Message:**
You are a **summarizer assistant** in an Image Quality Assessment (IQA) agent system. Your task is to integrate information from prior distortion analysis and computed IQA tool scores to produce a comprehensive quality interpretation and directly answer the user query.

**User Query:** {query}

**Your Input Includes:**

1. **Distortion Analysis:** Severity, category, and explanation of detected distortions per region or globally.

2. **IQA Tool Scores:** Quality scores (range 1 to 5) assigned by specific IQA tools based on the distortions.

3. **Reference Type:** Either `Full-Reference` or `No-Reference`, guiding tool usage and score interpretation.

4. **Optional Prior Answer:** A previously generated explanation you may consider for additional reasoning.

5. **Optional Image Input:** You may also infer the answer from images and query directly.

**Return a valid JSON list in the following format:**

```
{
  "quality_reasoning": "<Summary of the reasoning process,
  combining distortions, severity, descriptions, and IQA
  scores>",
  "final_answer": "<Concise and direct response to the user
  query based on the full reasoning>"
}
```

**Guidelines:**

- In "quality_reasoning":
    - Summarize the key distortions and their visual impact.
    - Reference tool scores to support your conclusion.
    - Ensure the logic connecting observations and conclusions are clear and interpretable.
- In "final_answer":
    - Provide a direct and concise judgment regarding the user query.
    - Use natural and human-readable phrasing.
- Only return the JSON object. Do not include any markdown, commentary, or additional text.

## C  MORE DETAILS ON AGENTICIQA-EVAL

Figure 5 illustrates representative samples from each subtask in **AgenticIQA-Eval**, covering the planner, executor (distortions and tools), and summarizer components.

- **Planner.** These questions assess the model's ability to parse the query and extract relevant planning cues. For example, given the query *"What is the primary distortion affecting the boat clarity?"*, the planner must identify the explicitly mentioned object (*Boat*) from the candidate options.

- **Executor (Distortions).** These questions test distortion classification and severity estimation. In the example shown, the model must judge that the distortion severity is *Mild* based on visible degradation levels.

- **Executor (Tools).** These items evaluate the appropriateness of tool selection for a given task. For instance, the model is asked whether the tool NIMA, designed for aesthetic quality assessment, is suitable for evaluating image *sharpness*. The correct answer is *No*, indicating that semantic tool knowledge is required.

- **Summarizer.** These questions assess the model's ability to reason over intermediate perceptual evidence (e.g., $M_t$) and generate consistent decisions aligned with the provided analysis. In the sample, the summarizer is prompted with a detailed explanation of blur and brightness degradation, along with a tool score from TOPIQ, and must infer the underlying reason for object obscurity in the distorted image. The correct answer—*loss of detail due to out-of-focus artifacts*—requires integrating tool output, visual semantics, and distortion understanding.

**Annotation Protocol.** All questions are authored by domain experts familiar with IQA and multimodal evaluation. Each MCQ is independently verified by a second annotator to ensure label correctness and question clarity. In cases of disagreement, a third expert adjudicates. The answer options are carefully designed to be plausible yet discriminative, ensuring that performance reflects genuine reasoning rather than pattern matching.

We maintain a balanced distribution of question formats across What, How, Which, and Yes/No types. Furthermore, we ensure equal coverage across FR and NR scenarios, distortion types (*e.g.*, blur, noise, compression), and planning configurations. The full benchmark will be released with standardized evaluation scripts, including MCQ parsing, answer validation, and per-track scoring.

| Planner | Executor (Distortions) | Executor (Tools) | Summarizer |
|---|---|---|---|
| 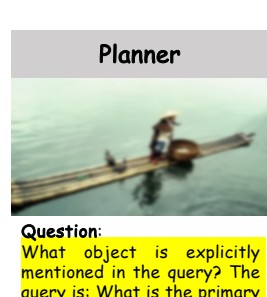 | 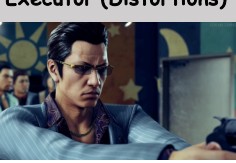 | | 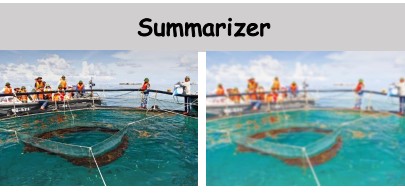 |

**Planner**

**Question**:
What object is explicitly mentioned in the query? The query is: What is the primary distortion affecting the boat clarity?
**Options**:
A. None
B. Boat
C. Person
D. Water
**Correct Answer**:
B. Boat

**Executor (Distortions)**

**Question**:
How severe is the distortion effect in the image?
**Options**:
A. None
B. Mild
C. Moderate
D. Severe
**Correct Answer**:
B. Mild

**Executor (Tools)**

**Question**:
Is the 'NIMA: Predicts aesthetic and technical quality using probability distributions derived from human ratings. Widely used for aesthetic assessment tasks.' tool appropriate for evaluating sharpness in the image?",
**Options**:
Yes,
No
**Correct Answer**: No

**Summarizer**

**Question**:
The following reasoning is provided about the image quality:\nThe image exhibits a moderate level of blur, which makes details less clear and the overall image soft. Additionally, the image appears brighter, affecting the contrast and potentially washing out some details.\n\nThe tool used is: TOPIQ_FR (score: 1.5963). The score ranges from 1 to 5, where a higher score indicates better quality.\nTOPIQ_FR is suitable because it excels at assessing blur and brightness changes, both of which are present here, by focusing on perceptually important distortion regions.\n\nBased on all the above information, please answer the following question:\nWhy might the details of the people and the water appear more obscure in the distorted image?
**Options**:
A. The distorted image is out of focus, causing a loss of detail
B. The color saturation is enhanced in the distorted image
C. The reference image has more visual clutter
D. The distorted image has too much brightness.
**Correct Answer**:
A. The distorted image is out of focus, causing a loss of detail

Figure 5: Illustrative examples from the AgenticIQA-Eval benchmark. Each subfigure corresponds to one evaluation component: (**Left** to **Right**) planner reasoning, distortion severity assessment, tool appropriateness, and summarization over multimodal evidence.

## D    QUALITATIVE VISUALIZATIONS

We present illustrative examples demonstrating the qualitative capabilities of the proposed AgenticIQA framework across both interpretation and scoring tasks. Specifically, visual results shown in Figs 6, 7, 8, and 9 highlight the system's effectiveness in generating accurate quality descriptions, identifying perceptually meaningful distortions, and providing coherent quality assessments aligned with human judgments.

## E    LIMITATIONS AND BROADER IMPACTS

### E.1    LIMITATIONS

While **AgenticIQA** demonstrates strong performance and interpretability, several limitations remain. First, the system assumes reliable task decomposition and tool selection by the planner; errors in early stages can propagate and degrade final outputs. Second, although we fine-tune an open-source VLM (Qwen2.5-VL), performance is still bounded by the language and vision capabilities of the backbone. Finally, while AgenticIQA is designed to be flexible, its execution speed and scalability may be constrained in real-time or resource-limited deployment settings due to the sequential nature of agentic reasoning.

### E.2    BROADER IMPACTS

AgenticIQA advances the field of perceptual quality assessment by enhancing transparency, adaptability, and multimodal alignment in IQA systems. Its ability to generate human-aligned explanations may facilitate fairer evaluation pipelines in applications such as generative media quality control, photo curation, and visual system benchmarking. However, like other VLM-based systems, AgenticIQA may inherit biases from training data or underlying models, potentially amplifying subjective quality judgments across demographic or cultural contexts. We encourage future work to further explore fairness, robustness under distribution shifts, and efficient inference to ensure responsible and equitable deployment.

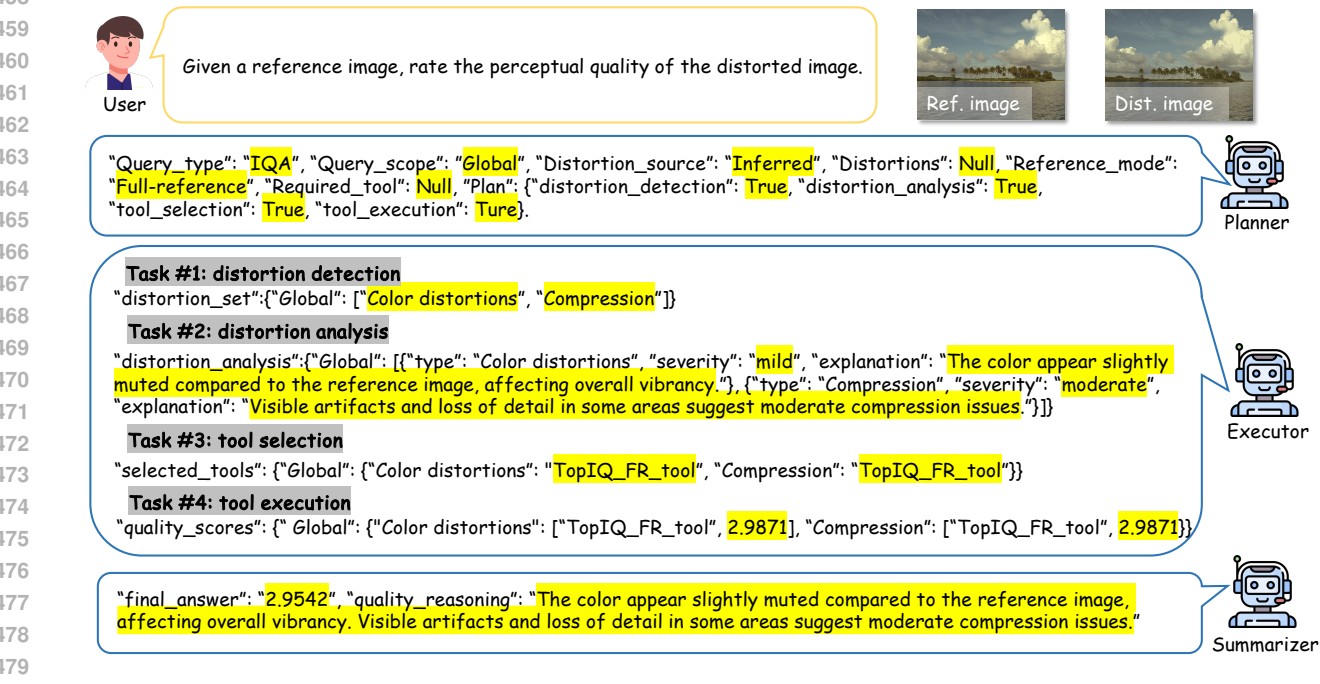

Figure 6: Qualitative result of AgenticIQA on the image quality scoring task.

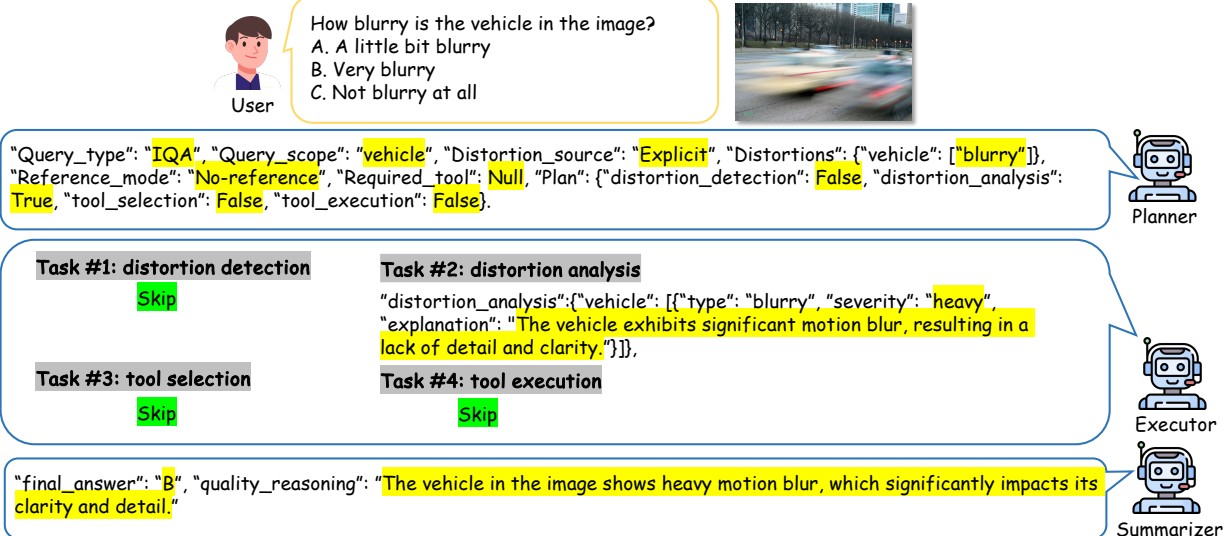

Figure 7: Qualitative result of AgenticIQA on the image quality description task.

## F DECLARATION OF LLM UTILIZATION

Large language models (LLMs) were used solely for minor language polishing and stylistic refinement of the manuscript. In dataset and benchmark construction, LLMs served as auxiliary tools for task generation and preliminary screening, whereas the overall framework design, data curation, and final validation were conducted by human researchers. The authors assume full responsibility for the integrity and accuracy of all content presented in this work.

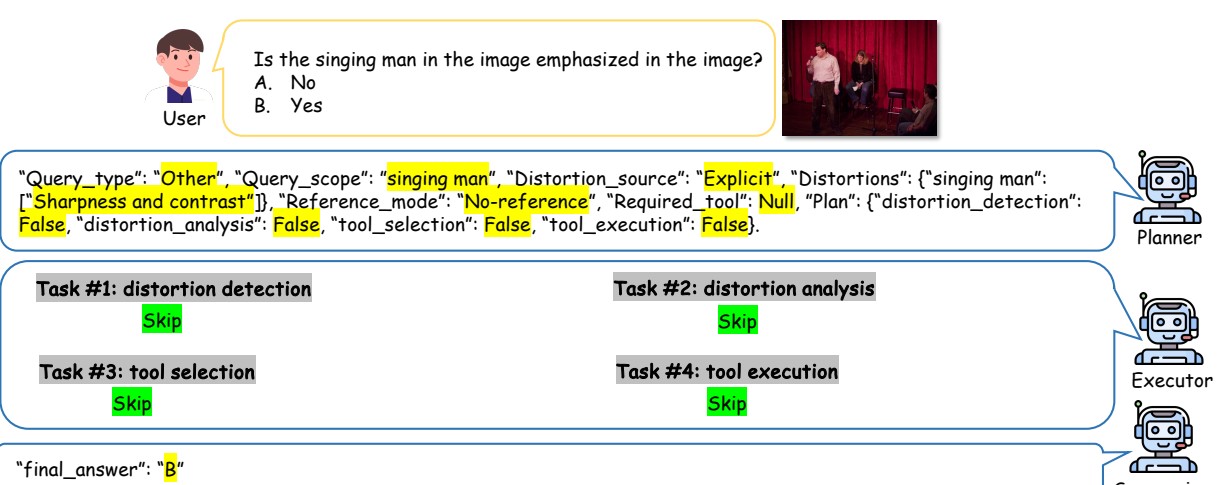

Figure 8: Qualitative result of AgenticIQA on the image quality description task.

**User:** What is the major distortion of this image?
A. Overexposure
B. Underexposure
C. Motion blur

**Planner:** "Query_type": "IQA", "Query_scope": "Global", "Distortion_source": "Explicit", "Distortions": {"Global": [" Motion blur", "Overexposure", "Underexposure"]}, "Reference_mode": "No-reference", "Required_tool": Null, "Plan": {"distortion_detection": False, "distortion_analysis": True, "tool_selection": True, "tool_execution": Ture}.

**Executor:**

Task #1: distortion detection
Skip

Task #2: distortion analysis
"distortion_analysis":{"Global": [{"type": "Overexposure", "severity": "none", "explanation": "There is no overexposure."}, {"type": "Underexposure", "severity": "heavy", "explanation": "The overall image appears significantly dark, obscuring details and reducing visibility."}, {"type": "Motion blur", "severity": "mild", "explanation": " There is some noticeable motion blur affecting the clarity of objects, particularly around moving elements."}]},

Task #3: tool selection
"selected_tools": {"Global": {"Overexposure": "QAlign_tool", "Underexposure": "QAlign_tool", "Motion blur": "QAlign_tool"}}

Task #4: tool execution
"quality_scores": {" Global": {"Overexposure": ["QAlign_tool", 2.6641], {"Underexposure": ["QAlign_tool", 2.6641], {"Motion blur": ["QAlign_tool", 2.6641]}

**Summarizer:** "final_answer": "B", "quality_reasoning": " The image exhibits under-exposure, resulting in significant darkness that obscures details and reduces visibility. This aligns with the distortion analysis indicating severe under-exposure as the primary issue."

**User:** Is the singing man in the image emphasized in the image?
A.  No
B.  Yes

**Planner:** "Query_type": "Other", "Query_scope": "singing man", "Distortion_source": "Explicit", "Distortions": {"singing man": ["Sharpness and contrast"]}, "Reference_mode": "No-reference", "Required_tool": Null, "Plan": {"distortion_detection": False, "distortion_analysis": False, "tool_selection": False, "tool_execution": False}.

**Executor:**

Task #1: distortion detection
Skip

Task #2: distortion analysis
Skip

Task #3: tool selection
Skip

Task #4: tool execution
Skip

**Summarizer:** "final_answer": "B"

Figure 9: Qualitative result of AgenticIQA on the image quality description task.

