# OpenReview forum: "AgenticIQA: An Agentic Framework for Adaptive and Interpretable Image Quality Assessment"
_ICLR.cc/2026/Conference — ICLR 2026 Conference Withdrawn Submission_

### Official Review · Reviewer_2xhB · 2025-10-28

**Soundness:** 3
**Presentation:** 2
**Contribution:** 2
**Rating:** 2
**Confidence:** 5

**Summary:**

This paper introduces AgenticIQA, a modular agentic framework that integrates vision-language models (VLMs) with traditional image quality assessment (IQA) tools. The AgenticIQA framework consists of three agents: a planner, an executor, and a summarizer. The planner formulates task-specific strategies, the executor collects perceptual evidence via tool execution, and the summarizer integrates this evidence to produce an IQA score with an accompanying quality explanation. In addition, the authors introduce the AgenticIQA-200K training dataset, a large-scale instruction dataset tailored for IQA agents, as well as AgenticIQA-Eval, a benchmark for assessing the planning, execution, and summarization capabilities of IQA agents. In experiments, AgenticIQA trained on AgenticIQA-200K demonstrates moderate performance.

**Strengths:**

Using a continually advancing general-purpose baseline VLM to build an agentic workflow that outputs both IQA scores and their accompanying rationales is a very reasonable approach.
- A system that provides IQA scores together with explanations can be highly useful in real-world applications.
- Enabling the use of approximately 30 IQA metrics as tools increases the flexibility of the proposed method.

**Weaknesses:**

At a high level, the proposed AgenticIQA architecture appears reasonable, but there are many aspects in the architectural details and experimental results that are difficult to evaluate positively.

Regarding the distortion detection module
- The distortion types used ("Blurs", "Color distortions", "Compression", "Noise", "Brightness change", "Sharpness", "Contrast") may suffice for qualitative descriptions of image quality, but it has not been thoroughly validated whether they are appropriate enough for zero-shot IQA score prediction.

About the performance of IQA Agent (Table 1)
- The proposed Qwen2.5-VL* model trained on AgenticIQA-200K underperforms the baseline Qwen2.5-VL in the Executor-Tool. How should this be interpreted?
- Based on the table 1, one would expect the best performance by using InternVL2.5 as the Planner, Q-Instruct and Qwen2.5-VL as the Executor (for Distortion and Tool, respectively), and LLaVA-OneVision as the Summarizer. Is there a specific reason for not adopting this configuration and instead using a single model, Qwen2.5-VL, for the Planner–Executor–Summarizer?

About the performance of IQA Scoring (Table 2)
- A fair comparison is difficult because there is no zero-shot methods.
- Although the IQA score performance is low, it would be beneficial to leverage the strengths of the proposed method and present a quality reasoning as a qualitative evaluation.
- The absence of baseline Qwen2.5-VL results makes accurate assessment difficult (it appears in Tables 1 and 3).

About the performance of IQA Interpretation (Table 3)
- The proposed Qwen2.5-VL* shows only a 0.03% improvement over the baseline Qwen2.5-VL, which casts doubt on the effectiveness of the method.

**Questions:**

No questions.

---

### Official Review · Reviewer_1cHU · 2025-10-28

**Soundness:** 4
**Presentation:** 3
**Contribution:** 4
**Rating:** 8
**Confidence:** 4

**Summary:**

This paper introduces AgenticIQA, a novel agentic framework for image quality assessment (IQA) that combines the strengths of traditional score-based IQA methods and VLM-based approaches. AgenticIQA decomposes the IQA process into four subtasks (distortion detection, distortion analysis, tool selection, and tool execution) orchestrated by a planner, executor, and summarizer. The paper also introduces AgenticIQA-200K, a large-scale instruction dataset for training IQA agents, and AgenticIQA-Eval, a benchmark for evaluating VLM-based IQA agents. Experimental results demonstrate that AgenticIQA outperforms strong baselines in both scoring accuracy and explanatory alignment.

**Strengths:**

1. Framing IQA as a multi-agent reasoning task is innovative and intellectually coherent with current LLM agent trends.
2. AgenticIQA-200K and AgenticIQA-Eval are valuable community resources likely to promote reproducibility and future research.
3. Experiments span human comparison, multiple datasets (TID2013, BID, AGIQA-3K, LLVisionQA), and multiple backbones (GPT-4o, Qwen2.5-VL), demonstrating robustness of the proposed agentic framework.
4. The paper is generally well-written and organized. The structure and logic are clear, figures are illustrative, and tables are easy to interpret.

**Weaknesses:**

1. The ablation focuses mainly on score fusion (HVS weighting vs. uniform). It would be stronger to also report results when removing specific agents (e.g., planner only / executor only).
2. Figure 3 shows runtime but lacks comparison to single-pass VLM baselines in terms of efficiency/accuracy trade-off.
3. The paper focuses on the successes of AgenticIQA. A brief discussion of potential failure cases or limitations would be beneficial.

**Questions:**

In addition to the aforementioned weaknesses, I have two additional questions:

1. Line 304: "synthetically degrade them with one or two randomly sampled distortions following the protocol of (You et al., 2024a)" – Please describe the types of distortions used in this synthetic degradation process.
2. It would be helpful to include a baseline that executes all available tools on the input image and feeds their outputs into the VLM model for prediction.

---

### Official Review · Reviewer_YsSd · 2025-10-29

**Soundness:** 2
**Presentation:** 2
**Contribution:** 2
**Rating:** 2
**Confidence:** 5

**Summary:**

This paper introduces AgenticIQA, a novel framework for image quality assessment (IQA) that addresses several limitations in current approaches. Traditional IQA models rely on fixed models that produce scalar quality scores or language-driven models that lack precision. AgenticIQA aims to bridge this gap by integrating traditional IQA tools with vision-language models (VLMs), creating a dynamic, query-aware system that enhances both scoring accuracy and interpretability. The framework operates through three components: planner, executor, and summarizer, which coordinate to detect and analyze distortions, select and apply IQA tools, and provide explanations.

**Strengths:**

1. The paper presents the AgenticIQA framework, which combines traditional image quality assessment methods with vision-language models (VLMs) that allow the system to adapt to different queries and images.
2. The paper introduces the AgenticIQA-200K dataset for training VLMs and the AgenticIQA-Eval benchmark.
3. The modular design of AgenticIQA allows independent handling of tasks (distortion detection, analysis, tool selection, etc.), offering greater flexibility and scalability compared to traditional integrated systems.

**Weaknesses:**

1. The AgenticIQA system requires significant computational resources due to its use of multiple agents and complex tasks, which may limit its efficiency in practical scenarios.
2. From the performance of IQA Scoring compared to baseline IQA methods, its performance is somewhat lower than that of other methods.
3. The paper does not conduct separate ablation studies on the individual modules of AgenticIQA (planner, executor, and summarizer), failing to fully verify the contribution of each module to the overall performance.
4. While the ablation studies demonstrate the framework's advantages, testing on complex distortion types and real-world scenarios is insufficient.
5. Besides, the submission format clearly does not meet the ICLR submission requirements, where the margin is too narrow.

**Questions:**

1. For the performance of AgenticIQA in Table 2, it shows lower results on some datasets compared to its counterpart methods. Please clarify it.
2. Others see weaknesses.

---

### Official Review · Reviewer_GY9x · 2025-10-31

**Soundness:** 2
**Presentation:** 2
**Contribution:** 2
**Rating:** 4
**Confidence:** 4

**Summary:**

The paper introduces AgenticIQA, a modular agent-based framework for Image Quality Assessment (IQA) that integrates traditional perceptual models with Vision-Language Models (VLMs) in a planner–executor–summarizer architecture. The approach decomposes the IQA process into subtasks (distortion detection, analysis, tool selection, and execution), coordinated by agents that perform reasoning and planning dynamically. The authors also release AgenticIQA-200K, a dataset designed for training and aligning VLMs with IQA-specific reasoning, and AgenticIQA-Eval, a benchmark to evaluate such systems. Experiments show that the framework outperforms both traditional IQA methods and VLM-based systems in accuracy and interpretability.

**Strengths:**

1. Novel framing – The idea of applying an “agentic” planning-execution-summarization paradigm to IQA is conceptually interesting and relatively unexplored. It takes cues from recent trends in LLM-based agents and adapts them to a low-level visual assessment domain, which is novel.

2. Comprehensive setup – The work is well-engineered: the framework, dataset, and benchmark together form a coherent ecosystem. The paper details how agentic reasoning helps modularize the IQA pipeline, which may inspire further work in interpretable assessment tasks.

3. Strong baselines and evaluation – The experiments are extensive, comparing with both classical (SSIM, LPIPS, DISTS, etc.) and recent VLM-based systems (Q-Instruct, Q-SiT). The authors also evaluate both the agentic reasoning ability (via multiple-choice benchmarks) and the final IQA accuracy.

4. Interpretability – Producing explicit, human-readable explanations is a clear plus, and aligns with the growing emphasis on transparency in perceptual quality tasks.

5. Dataset contribution – AgenticIQA-200K could be a useful community resource, especially if released with clear licensing and reproducibility.

**Weaknesses:**

1. Overly complex for the task – While the agentic formulation is creative, IQA is arguably a simpler regression problem compared to domains that benefit from multi-agent reasoning. The planner–executor–summarizer decomposition might feel over-engineered for predicting perceptual scores, adding unnecessary complexity without clear evidence that planning itself (rather than more data or supervision) provides the gains.

2. Limited interpretability validation – The claim of “interpretable” output is mostly qualitative. There is no human study to confirm that the generated explanations are actually more understandable or faithful to perceptual reasoning.

3. Computation overhead – The multi-agent design increases latency and resource usage substantially (as the paper admits). This could make practical deployment difficult for real-world systems that require fast, consistent predictions.

4. Ablations are incomplete – The ablation analysis is rather shallow. For example, it’s unclear how much each agent contributes to the overall gain. Would a single monolithic model trained on the same data achieve similar results?

5. Marginal quantitative improvement – In some datasets (e.g., AGIQA-3K), the improvement over baselines is modest. The additional architecture and training effort may not be justified by the relatively small gains.

6. Data generation pipeline ambiguity – The AgenticIQA-200K dataset appears to rely heavily on synthetic instructions and outputs generated via GPT-4o. This raises concerns about data originality, annotation quality, and possible circularity in evaluation (since GPT-4o is also used in the experiments).

**Questions:**

1. How sensitive is the system to the choice of underlying VLM backbone? Does the agentic design generalize if the base model is weaker (e.g., <7B models)?

2. Are the explanations consistent and faithful, or are they post-hoc justifications?

3. Can the agentic decomposition be learned end-to-end rather than hand-coded into planner/executor roles?

4. How does the system handle conflicting cues from different IQA tools?

---

### Note · Authors · 2025-11-16

I have read and agree with the venue's withdrawal policy on behalf of myself and my co-authors.